# Fast Path Planning for Kinematic Smoothing of Robotic Manipulator Motion

**DOI:** 10.3390/s25175598

**Published:** 2025-09-08

**Authors:** Hui Liu, Yunfan Li, Zhaofeng Yang, Yue Shen

**Affiliations:** School of Electrical and Information Engineering, Jiangsu University, Zhenjiang 212013, China; amity@ujs.edu.cn (H.L.); 2222307075@stmail.ujs.edu.cn (Y.L.); yangzhaofeng@stmail.ujs.edu.cn (Z.Y.)

**Keywords:** path planning, RRT*, artificial potential field, s-curve acceleration/deceleration

## Abstract

The Rapidly-exploring Random Tree Star (RRT*) algorithm is widely applied in robotic manipulator path planning, yet it does not directly consider motion control, where abrupt changes may cause shocks and vibrations, reducing accuracy and stability. To overcome this limitation, this paper proposes the Kinematically Smoothed, dynamically Biased Bidirectional Potential-guided RRT* (KSBB-P-RRT*) algorithm, which unifies path planning and motion control and introduces three main innovations. First, a fast path search strategy on the basis of Bi-RRT* integrates adaptive sampling and steering to accelerate exploration and improve efficiency. Second, a triangle-inequality-based optimization reduces redundant waypoints and lowers path cost. Third, a kinematically constrained smoothing strategy adapts a Jerk-Continuous S-Curve scheme to generate smooth and executable trajectories, thereby integrating path planning with motion control. Simulations in four environments show that KSBB-P-RRT* achieves at least 30% reduction in planning time and at least 3% reduction in path cost, while also requiring fewer iterations compared with Bi-RRT*, confirming its effectiveness and suitability for complex and precision-demanding applications such as agricultural robotics.

## 1. Introduction

Path planning [1,2] and motion control [3,4] are widely recognized as the two fundamental components of autonomous robotic manipulator systems, forming the basis for safe, stable, and precise task execution. With the continuous progress of intelligent robotics, application scenarios have become increasingly complex and diversified [5,6], placing higher demands on both planning efficiency and control robustness. Among these, agricultural environments represent a particularly challenging domain due to their dynamic, unstructured, and highly variable operating conditions [7,8]. These challenges underscore the critical importance of effective path planning and reliable motion control as prerequisites for achieving accurate and efficient manipulator performance in real-world applications.

Path planning is responsible for generating collision-free trajectories from an initial to a goal position [9], with its benefits clearly demonstrated in agricultural applications. Guan [10] reported that an optimized planner increased the picking success rate to 90%. For orchard harvesting, Liu [11] showed that an advanced planning approach improved harvesting efficiency and reduced task execution time in cluttered environments. Likewise, in a dual-arm fruit-harvesting system, Yoshida [12] demonstrated that an enhanced planner improved operational stability and reliability during harvesting operations, supporting its applicability under real field conditions.

These practical outcomes are underpinned by advances in robotic path planning algorithms, which are based on obstacle-avoidance requirements [13], such as Dijkstra’s algorithm [14], the A* algorithm [15], and the Rapidly-exploring Random Tree (RRT) algorithm [16]. Compared to graph-based algorithms like Dijkstra’s and A*, the RRT algorithm offers more efficient exploration of the configuration space and better adaptability to high-dimensional planning problems, which has led to its widespread application in robotic manipulator path planning. However, the standard RRT algorithm exhibits notable limitations, such as low path quality and slow planning speed, which hinder its ability to meet the stringent demands of real-time performance and robustness in complex environments. To address these limitations, various improved approaches have been proposed to enhance the performance of the RRT algorithm. One of the most influential variants is the RRT* algorithm [17], which improves the original RRT by reducing redundancy and enhancing path optimality. The RRT* employs an optimization strategy based on a minimum-cost path criterion, wherein parent nodes are selected from neighboring vertices to minimize cost, and a rewiring process is conducted to incrementally refine the search tree toward optimality, thereby significantly improving path quality. To reduce computational overhead, Lian [18] proposed the K-dimensional RRT (KD-RRT) algorithm, which utilizes a K-dimensional tree data structure to decrease the number of distance calculations during nearest-neighbor queries. Based on the concept of bidirectional expansion in RRT-Connect, Wang [19] introduced an improved Bidirectional RRT* (Bi-RRT*) algorithm by integrating the principles of Quick-RRT and RRT-Connect, thus accelerating convergence through faster bidirectional tree growth and improved search efficiency. Qiu [20] developed the Dynamic Bridging RRT algorithm, which enhances the search performance of RRT-Connect, thereby enabling safer and more efficient path navigation. Given RRT’s reliance on random sampling, Zhang [21] improved the Bias-RRT by incorporating probabilistic weighting factors into the sampling strategy, effectively reducing randomness and improving planning efficiency. Similarly, Gammell [22] proposed the Informed-RRT* algorithm, which restricts sampling to a progressively shrinking ellipsoidal region defined by the current best path, thereby focusing exploration and reducing path cost. To enhance guidance in complex environments, many improved RRT-based algorithms have incorporated Artificial Potential Field (APF) methods, where attractive forces guide the search toward the goal and repulsive forces prevent collision with obstacles. Yi [23] and Diao [24] integrated APF into their improved Potential-guided RRT* (P-RRT*) and APF-Improved RRT* (APF-IRRT*) algorithms, respectively, demonstrating enhanced adaptability in cluttered environments. Furthermore, Guo [25] proposed the Bidirectional P-RRT* algorithm with adaptive Direction-Biased and Variable-Step (DBVSB-P-RRT*), which integrates multiple advancements, including an adaptive deflection sampling mechanism, an improved formulation of the attractive force in the APF, and a variable step-size heuristic expansion strategy. These enhancements collectively contribute to faster path search and improved path quality.

Motion control ensures accurate joint actuation along planned trajectories [26], enabling smooth tracking and precise end-effector positioning. Its importance has been widely demonstrated in agricultural applications. In greenhouse seedling transplanting, Han [27] reported that advanced control significantly improved accuracy and efficiency. In apple harvesting, Chen [28] showed that force-sensitive strategies effectively reduced fruit damage. In vineyard spraying, Vatavuk [29] found that advanced control enhanced spray uniformity and reduced chemical waste under varying canopy conditions.

Collectively, these studies confirm that robust motion control directly improves the reliability and efficiency of robotic manipulators in real field operations, and it constitutes an essential stage following path planning by generating smooth and executable trajectories under motion constraints [30]. Such constraints enable manipulators to follow planned paths with minimal vibration [31,32], thereby enhancing mechanical durability and extending system lifespan. Several studies have incorporated such constraints into RRT-based algorithms to generate smoother trajectories. Webb [33] proposed the Kinodynamic RRT* algorithm, which integrates dynamic constraints to achieve asymptotically optimal motion planning for robots with linear differential constraints. Kang [34] introduced the Smooth-RRT* algorithm, which considers kinodynamic constraints during the RRT* rewiring process and thereby achieves shorter and smoother trajectories. Liu [35] proposed a Dubins-based RRT variant that optimizes path cost by applying post-processing waypoint shifts, significantly improving path smoothness and efficiency under kinematic constraints. Considering the kinematic characteristics of robotic manipulators, smooth motion control can also be achieved through explicit velocity, acceleration, and jerk constraints. Accordingly, Acceleration/Deceleration (ACC/DEC) algorithms, though traditionally applied in motor control, can be adapted to interpolate trajectories for end-effector motion. The Trapezoid ACC/DEC algorithm [36], though computationally simple, introduces discontinuities in acceleration, often resulting in dynamic shock during transitions. To mitigate this issue, the S-Curve ACC/DEC algorithm [37] introduces a transition phase in the acceleration profile, significantly reducing mechanical impact and producing smoother motion. As a further refinement, the Jerk-Continuous S-Curve ACC/DEC algorithm [38] has been proposed to eliminate jerk discontinuities and reduce flexible impacts caused by abrupt variations in jerk during movement transitions.

The reviewed studies have contributed to notable advancements in path planning and motion control, but several challenges and limitations still persist. In the domain of path search, although Bi-RRT* improves space exploration efficiency, it still lacks sufficient guidance in complex environments. Bias-P-RRT* enhances the target guidance mechanism to improve convergence rates, yet it remains limited in handling obstacle avoidance effectively. With respect to path optimization, the rewiring mechanism in RRT* reduces overall path cost, but redundant nodes still exist. Although the path pruning strategy can eliminate some of this redundancy, further optimization is still possible. In terms of trajectory smoothing, the Jerk-Continuous S-Curve ACC/DEC algorithm enables smooth and stable motion control, but adaptation is required for its effective application in trajectory smoothing.

In response to these challenges, this study proposes the Kinematically Smoothed, dynamically Biased Bidirectional Potential-guided RRT* (KSBB-P-RRT*) algorithm, which performs path planning under kinematic constraints while achieving shorter planning time, lower path cost, and reduced number of iterations. This makes it well suited for fine-grained and precision-demanding applications in robotic manipulators. The main contributions of this paper are summarized as follows:Fast Path Search Strategy: A multi-faceted enhanced search strategy is developed on the basis of Bi-RRT*, where a Dynamic Bias-Sample (DB-Sample) refines sampling guidance and an Adaptive P-Steer (AP-Steer) strengthens obstacle avoidance. Their integration accelerates bidirectional exploration and significantly improves search efficiency.Multi-step Triangle Optimization (MT-Optimize) Strategy: A path optimization strategy, designed on the basis of the triangle inequality, removes redundant waypoints and systematically adjusts remaining nodes, thereby refining the planned path and reducing overall path cost.Kinematically Constrained Smoothing (KC-Smooth) Strategy: A smoothing strategy is designed that incorporates the adaptation of the Jerk-Continuous S-Curve ACC/DEC algorithm to realize trajectory smoothing. This strategy generates a series of continuous trajectory points that are spatially smooth and temporally executable under kinematic constraints, thereby achieving the integration of path planning and motion control.

Figure 1 presents the general block diagram of this study, which illustrates the overall framework of the proposed approach integrating path search, optimization, and smoothing to generate executable trajectories for robotic manipulators. The remainder of this paper is organized as follows. Section 2 reviews the background algorithms, including Bi-RRT* and S-Curve ACC/DEC methods. Section 3 details the proposed KSBB-P-RRT* algorithm and its three key strategies. Section 4 reports comparative experiments against Bi-RRT* and Bias-P-RRT*, and further evaluates the improvements brought by the proposed path optimization and smoothing strategies. Finally, Section 5 summarizes the contributions and outlines future work.

## 2. Preliminaries

### 2.1. Bi-RRT* Algorithm

We define the configuration space as a set *X* [16], where $X_{\mathrm{space}}$ denotes the entire search space, $X_{\mathrm{obs}}$ the obstacle space, and $X_{\mathrm{free}}=X_{\mathrm{space}}\setminus X_{\mathrm{obs}}$ the obstacle-free space. A configuration x∈X represents a point in the configuration space, where $x_{\mathrm{init}}\in X_{\mathrm{free}}$ and $x_{\mathrm{goal}}\in X_{\mathrm{free}}$ denote the initial and goal points, respectively. Within *X*, an RRT is represented as T=(V,E), where *V* is the set of all nodes in *T*, and *E* is the set of edges connecting each node in *V* to its parent.

The Bi-RRT* algorithm [19], used for path planning, performs path search using a bidirectional tree strategy. The algorithm initializes two trees, $T_{1}$ and $T_{2}$, with the initial point $x_{\mathrm{init}}$ and the goal point $x_{\mathrm{goal}}$ as their respective root nodes. Subsequently, the algorithm alternates between the two trees, sequentially executing sampling and expansion, until a valid connection is formed between $T_{1}$ and $T_{2}$, leading to a complete path solution. This bidirectional strategy improves space exploration efficiency and accelerates convergence. However, due to its reliance on purely random sampling, the expansion process lacks guidance and exhibits excessive randomness.

The overall pseudocode [19] of the algorithm is presented in Algorithm 1. The required input parameters include the initial point $x_{\mathrm{init}}$, the goal point $x_{\mathrm{goal}}$, the search space $X_{\mathrm{space}}$, and the upper limit of iterations Iter. Explanations of the functions used in the pseudocode are provided below.

Sample: Returns a random sampling point $x_{\mathrm{rand}}$ in the search space.Nearest: Returns the nearest node $x_{\mathrm{nearest}}$ in the tree to the random point $x_{\mathrm{rand}}$.Steer: Returns a new node by moving a fixed step $d_{\mathrm{step}}$ from $x_{\mathrm{nearest}}$ towards $x_{\mathrm{rand}}$.CollisionFree: Checks whether the path between two nodes is free of collision.Near: Returns a set of nodes $X_{\mathrm{near}}$ near the new node $x_{\mathrm{new}}$ within a given radius or number.ChooseParent: Returns the optimal parent node for $x_{\mathrm{new}}$ from $X_{\mathrm{near}}\cup \left\{x_{\mathrm{nearest}}\right\}$ that yields the lowest path cost while ensuring collision-free connectivity.Rewire: Updates the parent nodes of nodes in $X_{\mathrm{near}}$ by evaluating whether connecting through $x_{\mathrm{new}}$ reduces their individual path costs.Swap: Swaps the two trees.

**Algorithm 1** Bi-RRT*($x_{\mathrm{init}},x_{\mathrm{goal}},X_{\mathrm{space}},Iter$)
1:

V1←{xinit},V2←{xgoal}

2:

E1←∅,E2←∅

3:

T1←(V1,E1),T2←(V2,E2)

4:**for** i=1 to Iter **do**5:    $x_{\mathrm{rand}}\leftarrow \mathrm{Sample}\left(i\right)$6:    $x_{\mathrm{nearest}}\leftarrow \mathrm{Nearest}\left(x_{\mathrm{rand}},T_{1}\right)$7:    $x_{\mathrm{new}}\leftarrow \mathrm{Steer}\left(x_{\mathrm{nearest}},x_{\mathrm{rand}},d_{\mathrm{step}}\right)$8:    **if** $\mathrm{CollisionFree}(x_{\mathrm{nearest}},x_{\mathrm{new}},X_{\mathrm{space}})$ **then**9:        $X_{\mathrm{near}}\leftarrow \mathrm{Near}\left(x_{\mathrm{new}},T_{1}\right)$10:        $x_{\mathrm{parent}}\leftarrow \mathrm{ChooseParent}\left(x_{\mathrm{new}},X_{\mathrm{near}},x_{\mathrm{nearest}}\right)$11:        $V_{1}\leftarrow V_{1}\cup \left\{x_{\mathrm{new}}\right\}$12:        $E_{1}\leftarrow E_{1}\cup \left\{\left(x_{\mathrm{new}},x_{\mathrm{parent}}\right)\right\}$13:        $T_{1}\leftarrow \mathrm{Rewire}\left(T_{1},x_{\mathrm{new}},X_{\mathrm{near}}\right)$14:        $x_{\mathrm{nearest}2}\leftarrow \mathrm{Nearest}\left(x_{\mathrm{new}},T_{2}\right)$15:        **if** $\mathrm{CollisionFree}(x_{\mathrm{nearest}2},x_{\mathrm{new}},X_{\mathrm{space}})$ and $∥x_{\mathrm{new}}-x_{\mathrm{nearest}2}{∥}_{2}<\delta$ **then**16:           **return** *T*17:        **end if**18:        $\mathrm{Swap}(T_{1},T_{2})$19:    **end if**20:
**end for**
21:**return** T=(V,E)


### 2.2. S-Curve ACC/DEC Algorithm

The S-Curve ACC/DEC algorithm is employed in motor control to generate executable trajectories that satisfy kinematic constraints. The algorithm consists of seven sequential phases, with the corresponding profiles of velocity, acceleration, and jerk (i.e., the rate of change of acceleration) illustrated in Figure 2, which is plotted by the authors for clarity based on the description in [39]. By enabling continuous variation in acceleration, the algorithm effectively eliminates sudden changes in acceleration, thereby reducing impact forces and achieving smooth motion control. However, a discontinuity still exists at the level of jerk, which can introduce flexible impacts and vibrations during motion transitions.

We define the following notations for the algorithm planning process. Let *s*, *v*, *a*, *j*, and *t* denote displacement, velocity, acceleration, jerk, and time, respectively. The maximum constraints are denoted as $v_{\max}$, $a_{\max}$, and $j_{\max}$, while $v_{s}$ and $v_{e}$ represent the initial and final velocities. Let $t_{i}$ (i=0,1,…,7) denote the time at each phase, $T_{i}=t_{i}-t_{i-1}$ the duration of the interval from $t_{i-1}$ to $t_{i}$, and $\tau_{i}=t-t_{i-1}$ the local time coordinate measured from $t_{i-1}$. The displacement, velocity, and acceleration at time $t_{i}$ are denoted as $S_{i}$, $V_{i}$, and $A_{i}$, respectively.

The algorithm imposes constraints on the maximum velocity $v_{\max}$, the maximum acceleration $a_{\max}$, and the maximum jerk $j_{\max}$, while also considering the start velocity $v_{s}$ and the end velocity $v_{e}$ for algorithm planning. The velocity profile can be obtained by integrating the acceleration, and the acceleration is defined as follows [39]:(1)$$a\left(t\right)=\left\{\begin{array}{cc} J\tau_{1}, & 0\leq t<t_{1}\\ A_{1}, & t_{1}\leq t<t_{2}\\ A_{1}-J\tau_{3}, & t_{2}\leq t<t_{3}\\ 0, & t_{3}\leq t<t_{4}\\ -J\tau_{5}, & t_{4}\leq t<t_{5}\\ A_{5}, & t_{5}\leq t<t_{6}\\ A_{5}+J\tau_{7}, & t_{6}\leq t<t_{7} \end{array}\right.$$
where $J=j_{\max}$, and $A_{i}$ denotes the acceleration at time $t_{i}$.

To complete the algorithm planning, it is necessary to compute the acceleration values $A_{1}$ and $A_{5}$, as well as the time durations $T_{i}$ for each phase. According to the characteristics of the algorithm, $T_{1}=T_{3}$ and $T_{5}=T_{7}$. Taking the acceleration phase ($t_{0}$ to $t_{3}$) as an example, the following relationships are satisfied [39]:(2)$$A_{1}=JT_{1}$$(3)$$V-v_{s}=A_{1}\left(T_{1}+T_{2}\right)$$
where *V* denotes the peak velocity. The expressions for computing $A_{1}$, $T_{1}$, and $T_{2}$ are derived by considering whether a constant acceleration phase is present or not, as follows [39]:(4)$$\left\{\begin{array}{cc} A_{1}=a_{\max},\;T_{1}=\frac{a_{\max}}{J},\;T_{2}=\frac{\Delta v}{a_{\max}}-\frac{a_{\max}}{J}, & \mathrm{if}\;\Delta v>\frac{a_{\max}^{2}}{J}\\ A_{1}=\sqrt{J\Delta v},\;T_{1}=\sqrt{\frac{\Delta v}{J}},\;T_{2}=0, & \mathrm{if}\;\Delta v\leq \frac{a_{\max}^{2}}{J} \end{array}\right.$$
where $\Delta v=|V-v_{s}|$. Similarly, for the deceleration phase ($t_{4}$ to $t_{7}$), based on the symmetry of the algorithm, the parameters $A_{5}$, $T_{5}$, and $T_{6}$ can be computed. At this stage, the peak velocity *V* and the duration of the constant velocity phase $T_{4}$ must be determined according to the total displacement $S_{t}$. The displacements during the acceleration phase $S_{a}=S_{3}-S_{0}$ and deceleration phase $S_{d}=S_{7}-S_{4}$ as well as the displacement constraint are computed as follows [39]:(5)$$S_{a}=\frac{1}{2}\left(v_{s}+V\right)\left(2T_{1}+T_{2}\right)$$(6)$$S_{d}=\frac{1}{2}\left(v_{e}+V\right)\left(2T_{5}+T_{6}\right)$$(7)$$S_{a}+S_{d}\leq S_{t}$$
Assuming $V=v_{\max}$, if the inequality in Equation (Equation 7) is satisfied, the algorithm planning can proceed as intended, and the remaining constant velocity duration $T_{4}$ is given by the following [39]:(8)$$T_{4}=\frac{S_{t}-S_{a}-S_{d}}{v_{\max}}$$
Otherwise, if the inequality is not satisfied, no constant velocity phase exists and $T_{4}=0$. In this case, the velocity *V* is progressively reduced using a bisection method or other numerical techniques until Equation (Equation 7) is satisfied, thus completing the planning process of the algorithm.

## 3. KSBB-P-RRT* Algorithm

The proposed KSBB-P-RRT* algorithm achieves shorter planning time, lower path cost, and reduced number of iterations, while enabling motion trajectory smoothing under kinematic constraints to realize efficient path planning and smooth motion control for robotic manipulators. The complete pseudocode of the KSBB-P-RRT* algorithm is presented in Algorithm 2, which was developed by the authors based on the framework of Algorithm 1. The modifications and additional function definitions introduced in this work are explained below.

DB_Sample: Returns a random sample from the search space based on the DB-Sample strategy, which adaptively selects a bias point with a probability that changes over iterations.AP_Steer: Returns a new node extended from $x_{\mathrm{nearest}}$ to $x_{\mathrm{rand}}$ using the AP-Steer strategy, which guides the extension under the influence of the goal point $x_{\mathrm{goal}}$.Optimize_Path: Returns an optimized path with lower cost using the MT-Optimize strategy, which removes redundant nodes and adjusts the positions of waypoints based on geometric heuristics.Smooth_Path: Returns a sequence of smoothed trajectory points using the KC-Smooth strategy, which ensures both spatial and temporal continuity of the trajectory.

**Algorithm 2** KSBB-P-RRT*($x_{\mathrm{init}},x_{\mathrm{goal}},X_{\mathrm{space}},Iter$)
1:

V1←{xinit},V2←{xgoal}

2:

E1←∅,E2←∅

3:

T1←(V1,E1),T2←(V2,E2)

4:**for** i=1 to Iter **do**5:    $x_{\mathrm{rand}}\leftarrow \mathrm{DB}\_\mathrm{Sample}\left(i\right)$6:    $x_{\mathrm{nearest}}\leftarrow \mathrm{Nearest}\left(x_{\mathrm{rand}},T_{1}\right)$7:    $x_{\mathrm{new}}\leftarrow \mathrm{AP}\_\mathrm{Steer}\left(x_{\mathrm{nearest}},x_{\mathrm{rand}},x_{\mathrm{goal}}\right)$8:    **if** $\mathrm{CollisionFree}(x_{\mathrm{nearest}},x_{\mathrm{new}},X_{\mathrm{space}})$ **then**9:        $X_{\mathrm{near}}\leftarrow \mathrm{Near}\left(x_{\mathrm{new}},T_{1}\right)$10:        $x_{\mathrm{parent}}\leftarrow \mathrm{ChooseParent}\left(x_{\mathrm{new}},X_{\mathrm{near}},x_{\mathrm{nearest}}\right)$11:        $V_{1}\leftarrow V_{1}\cup \left\{x_{\mathrm{new}}\right\}$12:        $E_{1}\leftarrow E_{1}\cup \left\{\left(x_{\mathrm{new}},x_{\mathrm{parent}}\right)\right\}$13:        $T_{1}\leftarrow \mathrm{Rewire}\left(T_{1},x_{\mathrm{new}},X_{\mathrm{near}}\right)$14:        $x_{\mathrm{nearest}2}\leftarrow \mathrm{Nearest}\left(x_{\mathrm{new}},T_{2}\right)$15:        **if** $\mathrm{CollisionFree}(x_{\mathrm{nearest}2},x_{\mathrm{new}},X_{\mathrm{space}})$ and $∥x_{\mathrm{new}}-x_{\mathrm{nearest}2}{∥}_{2}<\delta$ **then**16:           **return** *T*17:        **end if**18:        $\mathrm{Swap}(T_{1},T_{2})$19:    **end if**20:
**end for**
21:

$T\leftarrow \mathrm{Optimize}\_\mathrm{Path}(T_{1},T_{2})$

22:

T←Smooth_Path(T)

23:**return** T=(V,E)


In the path search stage, a Fast Path Search strategy is developed on the basis of the Bi-RRT* framework, with the integration of the DB-Sample strategy, which enhances sampling efficiency by reducing randomness through adaptive guidance, and the AP-Steer strategy, which adaptively adjusts the attractive coefficient and step size based on the environment. These enhancements to both the sampling and expansion strategies effectively realize a fast initial path search.

In the path optimization stage, the MT-Optimize strategy is proposed to refine the initial path by pruning redundant nodes and shortening the path length. The path optimization method trims each node and adjusts waypoints based on the triangle inequality principle, thereby reducing the path cost to enhance the overall quality of the path.

In the trajectory smoothing stage, the KC-Smooth strategy is proposed to generate a final path suitable for direct execution of the end-effector by robotic manipulators. This is achieved by employing an adapted Jerk-Continuous S-Curve ACC/DEC algorithm for trajectory interpolation, which ensures both spatial smoothing of path corners and temporal continuity of motion profiles.

### 3.1. Fast Path Search Strategy

#### 3.1.1. DB-Sample Strategy

The random sample strategy of conventional algorithms, while theoretically capable of covering the entire search space, often leads to inefficiencies due to the absence of directional guidance, particularly in complex environments. The Bias-Sample strategy [21] in Bias-RRT* improves this by introducing a constant probability of directly selecting the goal point, which helps reduce excessive randomness. However, its fixed nature limits adaptability during different stages of exploration. To address this issue, the DB-Sample strategy is proposed, which dynamically adjusts the biasing behavior over iterations. The pseudocode, written by the authors as an adaptation of the Bias-Sample method, is presented in Algorithm 3, and the key functions are explained as follows:BiasProbability: Returns the current probability of bias, dynamically computed on the basis of the current number of iterations.Rand: Returns a random number within the range [0, 1], used for probabilistic decision making.
**Algorithm 3** DB_Sample(*i*)1:$x_{\mathrm{sample}}\leftarrow \mathrm{Sample}\left(i\right)$2:$p_{\mathrm{bias}}\leftarrow \mathrm{BiasProbability}\left(i\right)$3:**if** $\mathrm{Rand}\left(0,1\right)\leq p_{\mathrm{bias}}$ **then**4:    **if** $\mathrm{Rand}\left(0,1\right)\leq p_{\mathrm{goal}}$ **then**5:        $x_{\mathrm{rand}}\leftarrow x_{\mathrm{goal}}$6:    **else**7:        $x_{\mathrm{rand}}\leftarrow \mathrm{Nearest}\left(x_{\mathrm{sample}},T_{2}\right)$8:    **end if**9:**else**10:    $x_{\mathrm{rand}}\leftarrow x_{\mathrm{sample}}$11:**end if**12:**return** $x_{\mathrm{rand}}$

The DB-Sample strategy improves on the conventional Bias-Sample strategy by introducing a dynamic probability $p_{\mathrm{bias}}$ to directly sample the dynamic bias point $x_{\mathrm{bias}}$ during random sampling. The overall biasing formulation is expressed as follows [21]:(9)$$x_{\mathrm{rand}}=\left\{\begin{array}{cc} x_{\mathrm{bias}}, & \mathrm{if}\;p_{\mathrm{rand}}\leq p_{\mathrm{bias}}\\ x_{\mathrm{sample}}, & \mathrm{if}\;p_{\mathrm{rand}}>p_{\mathrm{bias}} \end{array}\right.$$
where $x_{\mathrm{rand}}$ denotes the final sampled point, and $x_{\mathrm{sample}}$ is a uniformly random sample from the search space. A random probability $p_{\mathrm{rand}}\in \left[0,\;1\right]$ is generated for sampling decision. If $p_{\mathrm{rand}}\leq p_{\mathrm{bias}}$, the bias point $x_{\mathrm{bias}}$ is selected; otherwise, the sample point $x_{\mathrm{sample}}$ is chosen.

The dynamic probability of bias $p_{\mathrm{bias}}$ increases progressively with the number of iterations, thereby ensuring sufficient breadth and depth of exploration throughout the planning process. Specifically, during the early stages of iteration, when the tree contains relatively few nodes, a lower bias probability promotes greater sampling randomness, facilitating broad exploration of the search space. In contrast, during the later stages, when the tree has expanded considerably, the spatial coverage is typically sufficient. At this point, a higher bias probability improves directional guidance, enhancing local refinement and accelerating convergence. The formulation of the dynamic probability $p_{\mathrm{bias}}$, proposed in this work, is given as follows:(10)$$p_{\mathrm{bias}}=\left\{\begin{array}{cc} \frac{i}{i_{\max}}p_{\max}, & \mathrm{if}\;i\leq i_{\max}\\ p_{\max}, & \mathrm{if}\;i>i_{\max} \end{array}\right.$$
where $p_{\max}$ is the maximum probability of bias and must remain less than 1 to ensure adequate randomness. *i* is the current number of iterations, and $i_{\max}$ is the number of iterations at which $p_{\mathrm{bias}}$ reaches its maximum.

The selection of dynamic bias points $x_{\mathrm{bias}}$ is proposed as an enhanced strategy based on the Bi-RRT* framework. In this context, two trees are grown, one from the initial point $x_{\mathrm{init}}$ (denoted as $T_{1}$) and the other from the goal point $x_{\mathrm{goal}}$ (denoted as $T_{2}$). Unlike conventional approaches, where each tree selects its own goal as the bias point, the proposed strategy incorporates a fixed probability of selecting a node from the opposing tree as the bias point. This strategy increases the likelihood of a successful connection between the two trees. The selection rule for the bias point $x_{\mathrm{bias}}$, formulated in this study, is defined as follows:(11)$$x_{\mathrm{bias}}=\left\{\begin{array}{cc} x_{\mathrm{goal}}, & \mathrm{if}\;p_{\mathrm{rand}}^{\prime }\leq p_{\mathrm{goal}}\\ x_{\mathrm{opposite}}, & \mathrm{if}\;p_{\mathrm{rand}}^{\prime }>p_{\mathrm{goal}} \end{array}\right.$$
where $x_{\mathrm{goal}}$ refers to the target point of $T_{1}$ (for $T_{2}$, the target point is $x_{\mathrm{init}}$), $x_{\mathrm{opposite}}$ denotes the nearest existing node in the opposite tree to the current random sample $x_{\mathrm{sample}}$, and $p_{\mathrm{goal}}$ is the fixed probability of selecting the target point as the bias point.

#### 3.1.2. AP-Steer Strategy

Expansion in traditional algorithms typically proceeds toward random sample points using a fixed step size, which frequently results in excessive iterations and inadequate adaptability in complex environments. In response to this challenge, the P-Steer strategy [40] from the P-RRT* algorithm incorporates the APF method, employing the attractive force of the goal point to perform multi-step expansion and accelerate exploration. However, it lacks effective obstacle avoidance capabilities when encountering large obstacles. To overcome this limitation, this study proposes the AP-Steer strategy that dynamically adjusts the attractive force coefficient in the APF and adapts the expansion step size according to the environment, thus enhancing the adaptability of the algorithm. Algorithm 4 presents the pseudocode of the proposed strategy, which has been adapted from the original P-Steer method for improved applicability, followed by the description of the function.

NearestObstacle: Returns the obstacle nearest to $x_{\mathrm{new}}$, used for force computation in the APF.

**Algorithm 4** AP_Steer($x_{\mathrm{nearest}},x_{\mathrm{rand}},x_{\mathrm{goal}}$)
1:

$x_{\mathrm{new}}\leftarrow \mathrm{Steer}\left(x_{\mathrm{nearest}},x_{\mathrm{rand}},d_{\mathrm{step}}\right)$

2:**for** i←1 to Iter **do**3:    $D\left(x,x_{\mathrm{obs}}\right)\leftarrow \mathrm{NearestObstacle}\left(x_{\mathrm{new}}\right)$4:    **if** $D\left(x,x_{\mathrm{obs}}\right)\leq D_{\mathrm{obs}}$ **then**5:        $k_{a}\leftarrow \frac{D(x,x_{\mathrm{obs}})}{D(x,x_{\mathrm{goal}})}$6:        ${\vec{F}}_{t}\leftarrow k_{a}\left(x_{\mathrm{goal}}-x_{\mathrm{new}}\right)+\left(x_{\mathrm{new}}-x_{\mathrm{obs}}\right)$7:    **else**8:        ${\vec{F}}_{t}\leftarrow \left(x_{\mathrm{goal}}-x_{\mathrm{new}}\right)$9:    **end if**10:    $x_{\mathrm{new}}\leftarrow x_{\mathrm{new}}+d_{\lambda }\frac{{\vec{F}}_{t}}{∥{\vec{F}}_{t}{∥}_{2}}$11:
**end for**
12:**return** $x_{\mathrm{new}}$


The AP-Steer strategy extends the conventional extension toward $x_{\mathrm{rand}}$ with a fixed step size $d_{\mathrm{step}}$, and further performs a multi-step extension with a fixed step size $d_{\lambda }$ in the direction of the total force computed via the APF. The total force $F_{t}$ is composed of the attractive force $F_{a}$ from the goal $x_{\mathrm{goal}}$ and the repulsive force $F_{r}$ from obstacles. However, a fixed attractive coefficient $k_{a}$ is employed in the traditional APF method to guide the tree toward the goal, resulting in limited adaptability in complex environments. When using a fixed attractive coefficient during expansion, if the new node $x_{\mathrm{new}}$ moves into a region directly obstructed by obstacles, an excessively large coefficient can hinder $x_{\mathrm{new}}$ from successfully bypassing the obstacle. Conversely, if $x_{\mathrm{new}}$ enters a relatively open region but is still affected by repulsive forces, an insufficient attractive coefficient may slow the convergence toward the goal. Therefore, an adaptive dynamic attractive coefficient is proposed to accommodate the varying conditions encountered during the expansion process. The formulation builds upon the attractive force computation in [25], and is modified in this work as follows:(12)$$k_{a}=\left\{\begin{array}{cc} \frac{D(x,x_{\mathrm{obs}})}{D(x,x_{\mathrm{goal}})}, & D\left(x,x_{\mathrm{obs}}\right)\leq D_{\mathrm{obs}}\\ 1, & D\left(x,x_{\mathrm{obs}}\right)>D_{\mathrm{obs}} \end{array}\right.$$
where $D(x,x_{\mathrm{goal}})$ is the distance to the goal, $D(x,x_{\mathrm{obs}})$ is the distance to the nearest obstacle, and $D_{\mathrm{obs}}$ is the threshold within which repulsion is activated. When the new node $x_{\mathrm{new}}$ enters the repulsive field of an obstacle, the attractive coefficient decreases rapidly. The closer it is to the obstacle, the weaker the attractive force becomes. In such cases, the motion is primarily driven by the repulsive force, enabling the expansion to quickly move away from the obstacle region and thereby enhancing obstacle avoidance capability.

In environments with diverse search space scales and obstacle distributions, the fixed step size $d_{\mathrm{step}}$ employed in traditional algorithms proves inflexible, as it necessitates manual adjustment to suit specific conditions, resulting in increasing implementation complexity and reducing adaptability. Inspired by the dynamic step-size mechanism in [41], this study proposes an adaptive computation method for $d_{\mathrm{step}}$, which considers the range and dimensionality of the search space as well as the obstacle occupancy ratio, and is formulated as follows:(13)$$d_{\mathrm{step}}=\frac{1}{\gamma }D_{\mathrm{avg}}\sqrt{N_{\dim}}\left(1-R_{\mathrm{obs}}\right)$$
where γ is the step adjustment coefficient, $D_{\mathrm{avg}}$ is the average search range per dimension, $N_{\dim}$ is the number of dimensions, and $R_{\mathrm{obs}}$ is the ratio of obstacle area (or volume) to the total space.

### 3.2. MT-Optimize Strategy

Although the DB-Sample and AP-Steer strategies enhance planning speed, the resulting initial path still requires optimization to achieve higher quality. Consequently, the MT-Optimize strategy is proposed in this study, comprising three stages.

As the first stage, a pruning strategy [42] is employed to remove redundant nodes from the generated path, as shown in Figure 3. Starting from the initial point $x_{init}$, the strategy attempts to connect it sequentially with subsequent nodes along the path. When the line segment connecting $x_{init}$ to $x_{3}$ collides with an obstacle, the previous node $x_{2}$ is selected as the next path point, and the intermediate point $x_{1}$ is considered redundant and removed. This process is repeated iteratively by traversing all remaining path points to eliminate any redundant nodes along the trajectory.

As the second stage, this study proposes a path-shortening strategy based on the triangle inequality, applied after the initial path pruning. The objective is to further minimize the overall path cost by iteratively adjusting intermediate waypoints. Specifically, for each waypoint with two neighboring nodes, the point can be shifted within the triangle formed by these three nodes to potentially reduce the total path length. As illustrated in Figure 4, which is drawn by the authors for clarity, the optimization of a waypoint $x_{1}$, situated between $x_{0}$ and $x_{2}$, is carried out in three steps. First, in Figure 4a, a preprocessing step similar to the pruning strategy is applied. If the direct connection between $x_{0}$ and $x_{2}$ does not intersect any obstacles, then $x_{1}$ is deemed redundant and removed, and the remaining two steps are skipped. If a collision occurs, in Figure 4a, the second step proceeds by moving $x_{1}$ incrementally toward $x_{0}$ with a fixed minimum step size *d*, until the segment between the updated position and $x_{2}$ intersects with an obstacle. The last collision-free position is retained as $x_{1}^{\prime }$. Third, in Figure 4b, using a method similar to the second step, the intermediate point $x_{1}^{\prime }$ is further moved toward $x_{2}$ until a final optimized position $x_{1}^{\prime \prime }$ is obtained, which replaces the original waypoint $x_{1}$. This process is applied iteratively to all eligible waypoints in the path. Optimization terminates when no further redundant nodes exist and point positions converge. The resulting path tightly conforms to obstacle boundaries while achieving a lower overall path cost.

As the third stage, this study proposes a path refinement strategy to address potential issues introduced by the previous shortening process. Specifically, due to the influence of obstacle contours, the optimized path may occasionally yield adjacent waypoints that are overly close to each other, which adversely affects subsequent trajectory smoothing. To resolve this, a threshold distance $d_{\mathrm{limit}}$ is defined, and the path refinement process is illustrated by the authors in Figure 5. If the distance between two consecutive waypoints $x_{1}$ and $x_{2}$ is below the threshold $d_{\mathrm{limit}}$, a corrective step is performed. An intersection point $x_{\mathrm{new}}$ is computed between the lines defined by points $x_{0}$, $x_{1}$ and $x_{3}$, $x_{2}$, and this point replaces both $x_{1}$ and $x_{2}$. This extension-and-merge operation is applied iteratively along the entire path to eliminate overly close waypoints. Through this process, the overall path optimization is finalized, generating a higher-quality path. Although this correction may slightly increase the path cost, it provides additional planning space for subsequent smoothing, and thus the overall path quality can still be considered improved.

### 3.3. KC-Smooth Strategy

Traditional interpolation-based smoothing methods, such as cubic spline interpolation, may generate poorly constrained trajectories, which in turn increase the likelihood of collisions with obstacles. Moreover, such methods typically ignore physical constraints such as kinematics, which results in planned paths that are not directly executable and therefore require additional trajectory tracking algorithms to compensate for feasibility issues. In response to these limitations, this study proposes the KC-Smooth strategy, which integrates trajectory planning and control, allowing collision checking during smoothing and direct incorporation of kinematic constraints into the interpolation process. As a result, the generated trajectory points can be directly utilized as finely controlled end-effector motion commands for execution by the manipulator.

The KC-Smooth strategy consists of both spatial and temporal components. Spatially, corner regions around waypoints are smoothed using curve transitions, while the remaining segments retain straight-line paths, ensuring controllability of the smoothed trajectory. Temporally, constant velocity motion is maintained along linear segments, and smooth velocity transitions are achieved at the initial point, the goal point, and each corner via kinematic profile planning, promoting smooth and stable execution of the overall motion. To implement the KC-Smooth strategy, it is necessary to determine the positions and velocities at the start and end of each corner smoothing segment, as illustrated by the authors in Figure 6. An appropriately adapted ACC/DEC algorithm is proposed to perform trajectory planning at the corners.

A Jerk-Continuous S-Curve ACC/DEC algorithm [38] is employed to ensure smooth motion by mitigating the impact caused by abrupt acceleration changes and the flexible vibration induced by sudden jerk transitions. This algorithm replaces the linear acceleration phase of traditional S-Curve ACC/DEC with a cosine-based fitting function, yielding a continuous jerk profile. To apply this algorithm for corner smoothing, appropriate adaptations are required. Spatially, the original algorithm is designed for linear motion control; thus, its kinematic quantities must be reformulated as spatial vectors. Temporally, only the acceleration phase (from $t_{0}$ to $t_{3}$, as shown in Figure 2) is used to achieve smooth transitions between start and end velocity vectors at the corners. The adapted acceleration and jerk profiles are illustrated in Figure 7.

The adapted Jerk-Continuous S-Curve ACC/DEC algorithm imposes kinematic constraints on the maximum velocity $v_{\max}$, maximum acceleration $a_{\max}$, and maximum jerk $j_{\max}$. We define the bold symbols x, v, a, and j as the vector forms of position, velocity, acceleration, and jerk, respectively. Building on the formulation in [38], the acceleration vector a(t) is adapted in this study as follows:(14)$$a\left(t\right)=\left\{\begin{array}{cc} \frac{A}{2}\left(1-\cos\frac{\pi \tau_{1}}{T_{1}}\right), & 0\leq t<t_{1}\\ A, & t_{1}\leq t<t_{2}\\ \frac{A}{2}\left(1+\cos\frac{\pi \tau_{3}}{T_{1}}\right), & t_{2}\leq t<t_{3} \end{array}\right.$$
where A represents the peak acceleration vector. The corresponding velocity vector v(t) and position vector x(t) are derived by integrating a(t) over time [38]:(15)$$v\left(t\right)=v_{s}+\int_{0}^{t}a\left(t\right)\;dt,\;0\leq t<t_{3}$$(16)$$x\left(t\right)=x_{s}+\int_{0}^{t}v\left(t\right)\;dt,\;0\leq t<t_{3}$$
where $v_{s}$, $v_{e}$, and $x_{s}$, $x_{e}$ represent the velocity and position vectors at the start and end of the corner smoothing segment, respectively, within which the trajectory smoothing is performed. Based on equivalent calculations, the jerk vector in the S-Curve ACC/DEC algorithm is defined as $J=\frac{2}{\pi }j_{\max}$, and the peak velocity vector is given by $V=v_{e}$. Using these formulations, the peak acceleration vector A, the time intervals $T_{1}$, $T_{2}$, and the displacement from $x_{s}$ to $x_{e}$ can be computed via Equations (Equation 4) and (Equation 5). Ultimately, the problem is reduced to trajectory planning based on the start and end velocity vectors $v_{s}$ and $v_{e}$.

By applying the adapted Jerk-Continuous S-Curve ACC/DEC algorithm, the position and velocity vectors at the start and end of each corner smoothing segment are planned. This enables the completion of both path smoothing and motion trajectory generation. The detailed steps of the KC-Smooth strategy proposed in this study are described below.

Step 1: Preplan the start and end velocity vectors ($v_{si}$ and $v_{ei}$) corresponding to each corner point $x_{i}$ (i=1,2), as illustrated in Figure 6. The pre-planning begins by assuming a maximum velocity magnitude $v_{\max}$, which is then reduced iteratively based on the motion distance and collision constraints until all feasibility conditions are satisfied.

Step 2: Determine the constant velocity for each linear path segment. For a segment connecting $x_{1}$ and $x_{2}$ shown in Figure 6, the uniform velocity is defined as the vector with the smaller magnitude between the end velocity $v_{e1}$ of the preceding corner and the start velocity $v_{s2}$ of the subsequent corner. This approach is applied consistently across all segments to ensure smooth transitions.

Step 3: Perform trajectory point interpolation. Based on the planned constant velocities, the trajectory is smoothed by incrementally integrating over time using the Equations (Equation 14)–(Equation 16) from the adapted Jerk-Continuous S-Curve ACC/DEC algorithm. This process yields a continuous and kinematically feasible trajectory along the entire path.

## 4. Simulation and Results

Through simulation experiments, the proposed KSBB-P-RRT* algorithm is compared with the Bi-RRT* and Bias-P-RRT* algorithms to evaluate overall planning performance. In addition, the effectiveness of path optimization and trajectory smoothing is also assessed independently. The simulations are conducted on an Intel i9-13900HX CPU (Intel, Santa Clara, CA, USA) with 16 GB of RAM, running on an HP Omen 9 laptop (HP Inc., Palo Alto, CA, USA). All algorithms are implemented using Python 3.10.16.

### 4.1. Simulation Environment Setup

The proposed algorithms are tested in both 2D and 3D environments, each comprising two levels of complexity, simple and complex maps, as illustrated in Figure 8.

Figure 8a,b depict the 2D environments. The search space is defined as a 100×100 unit area, containing multiple black rectangles and circles representing obstacles. Each obstacle is inflated with a buffer radius of 3 units. The unit and goal points are marked by green and red dots, respectively, located at coordinates (5, 5) and (95, 95). Figure 8a represents a simple 2D environment with fewer but larger obstacles, while Figure 8b shows a more complex 2D environment with a greater number of smaller obstacles.

Figure 8c,d present the 3D environments. The search space spans 100×100×100 units, where obstacles are represented by semi-transparent green spheres, also inflated with a 3-unit buffer radius. The unit and goal points are positioned at coordinates (0, 0, 0) and (100, 100, 100), respectively. Figure 8c illustrates a simple 3D environment with fewer but larger obstacles, whereas Figure 8d corresponds to a complex 3D environment with a larger number of smaller obstacles.

### 4.2. Trajectory Planning Performance Analysis of RRT-Based Algorithms

To evaluate planning performance, the three algorithms Bi-RRT*, Bias-P-RRT*, and the proposed KSBB-P-RRT* were tested in the four aforementioned environments, producing 3 × 4 experimental groups. A successful trial was defined as the generation of a collision-free path from the initial position to the goal position. Performance was evaluated using four key metrics: planning time *T*, path cost *C*, number of iterations *I*, and success rate. To mitigate the influence of randomness, each algorithm was executed 100 times per scenario, with the average values of these performance metrics serving as the primary basis for comparison.

#### 4.2.1. Parameter Settings

The parameter settings are summarized in Table 1, where $d_{\mathrm{step}}$ denotes the initial step size. For Bi-RRT* and Bias-P-RRT*, the step size is fixed at 5 in 2D environments and 7 in 3D environments. In contrast, KSBB-P-RRT* employs an adaptive step size mechanism, where γ is the step size adjustment coefficient. Iter represents the upper limit of iterations, $N_{\mathrm{rewire}}$ is the maximum number of neighboring nodes considered during rewiring, $D_{\mathrm{obs}}$ denotes the obstacle repulsion distance, $d_{\lambda }$ represents the base step size used for multi-step extension, and $p_{\mathrm{bias}}$ is the probability of goal-biased sampling.

#### 4.2.2. Experimental Data

Based on 100 runs per scenario, the relevant performance metrics of the three algorithms across four benchmark environments (as illustrated in Figure 8) are summarized in Table 2. Here, $T_{\mathrm{avg}}$ represents the average planning time, $C_{\mathrm{avg}}$ denotes the average path cost, $I_{\mathrm{avg}}$ indicates the average number of iterations, and $I_{\max}$ corresponds to the maximum number of iterations observed among all trials. Due to the relatively high upper limit allowable for iterations, all three algorithms achieved a success rate of 100%. Therefore, the subsequent analysis focuses on the remaining three metrics.

#### 4.2.3. Data Comparison of Planning Time

In terms of planning time, as shown in the $T_{\mathrm{avg}}$ column of Table 2, Bi-RRT* exhibits moderate adaptability across all environments. However, in complex environments with dense obstacles, its lack of goal-directed guidance results in significantly longer planning times. Bias-P-RRT* incurs longer planning times in simple environments with large obstacles. This is mainly due to the overly strong attraction toward the goal, which makes it difficult for the planner to move away from nearby obstacles. In contrast, KSBB-P-RRT* integrates the strengths of both previous algorithms and shows consistently superior performance across all environments. Compared to Bi-RRT*, it reduces the average planning time by 36.85%, 33.37%, 55.91%, and 70.35% in the four environments. Compared to Bias-P-RRT*, the reductions are 57.14%, 18.83%, 84.07%, and 67.62%, respectively. These results demonstrate that KSBB-P-RRT* significantly improves planning efficiency, especially in 3D environments. This efficiency gain is primarily attributed to the use of the DB-Sample strategy, which facilitates more effective goal-biased sampling, and the AP-Steer strategy enables adaptive extension toward feasible directions. The reduction in planning time supports faster exploration and convergence, which is favorable for responsive and efficient motion planning.

#### 4.2.4. Data Comparison of Path Cost

In terms of path cost, as shown in the $C_{\mathrm{avg}}$ column of Table 2, the Bias-P-RRT* algorithm, which uses the goal as an attractive guide, generally produces lower path cost than Bi-RRT* across all environments. Building on this advantage, KSBB-P-RRT* further improves path quality by applying the MT-Optimize and KC-Smooth strategies. Compared to Bi-RRT*, it reduces the average path cost by 3.36%, 6.41%, 8.11%, and 6.27% across the four environments. Compared to Bias-P-RRT*, the reductions are 0.43%, 2.79%, 0.67%, and 0.72%. These results validate the superior path quality of KSBB-P-RRT* across diverse scenarios. This improvement is primarily due to the use of MT-Optimize, which removes redundant segments and reduces path tortuosity, and KC-Smooth, which smooths sharp corners along the path. Together, these strategies produce more concise and efficient trajectories, resulting in a lower overall path cost. Minimizing path cost is theoretically beneficial for reducing energy expenditure during execution, potentially supporting more efficient and reliable manipulator operation.

#### 4.2.5. Data Comparison of Number of Iterations

The number of iterations reflects both the success rate of expansion attempts and the adaptability of the algorithm to the environment, as indicated by the $I_{\mathrm{avg}}$ and $I_{\max}$ columns in Table 2. In the simple 3D environment, both Bi-RRT* and Bias-P-RRT* required over 200 iterations on average. According to the maximum number of iterations observed in all trials, it can be inferred that reducing the upper limit of iterations to 800 would cause the success rates of these two algorithms to fall below 100%. This is mainly because the large obstacles in the environment pose significant challenges: Bi-RRT*, despite its efficient sampling, lacks effective obstacle avoidance strategies, while Bias-P-RRT*, although incorporating the APF method, still struggles to navigate around large obstacles. These limitations result in frequent expansion failures and an increased number of iterations for both algorithms. In contrast, KSBB-P-RRT* maintains an average number of iterations below 100 across all environments and continues to achieve a high success rate even when the upper limit of iterations is relatively low. This performance is attributed to the use of the AP-Steer strategy, which adaptively adjusts the step size and attractive force coefficients based on the proximity and distribution of surrounding obstacles, thereby enhancing both search efficiency and obstacle avoidance capability. Reducing the number of iterations is expected to enhance planning stability and mitigate computational burden, especially in time-sensitive manipulation scenarios.

#### 4.2.6. Visualization Comparison of Planned Paths

To provide an intuitive comparison, Figure 9, Figure 10, Figure 11 and Figure 12 illustrate the planned paths of the three algorithms, with each figure showing a representative result selected from 100 runs in each scenario.

The red line indicates the final planned path, while the blue lines represent the branches of the search tree. Compared to Bi-RRT* and Bias-P-RRT*, the proposed KSBB-P-RRT* generates fewer redundant tree branches and produces smoother, more natural paths for execution, demonstrating enhanced convergence and path quality. This performance benefit results from the integration of the DB-Sample strategy for more goal-directed sampling, the AP-Steer strategy for adaptive tree extension, the MT-Optimize strategy for reducing path tortuosity, and the KC-Smooth strategy for smoothing sharp transitions along the path. The algorithm yields continuous and kinematically consistent trajectory points, promoting smooth and stable motion along the planned path.

### 4.3. Comparison of Optimization Performance at Sequential Planning Phases

The proposed algorithm incorporates the MT-Optimize and KC-Smooth strategies for path refinement. To evaluate the effectiveness of these optimization stages, a comparative analysis is performed. Specifically, the average results from the above 100-run experiments in each benchmark environment are used to compare the path costs at different optimization stages. Table 3 summarizes the average path costs at three optimization phases across the four environments. Here, $C_{\mathrm{init}}$ denotes the initial average path cost obtained using the DB-Sample and AP-Steer strategies; $C_{\mathrm{opt}}$ represents the average path cost after applying MT-Optimize; and $C_{\mathrm{smooth}}$ corresponds to the final average path cost after further applying KC-Smooth.

The MT-Optimize strategy eliminates redundant waypoints from the initial path and adjusts remaining waypoints toward directions with lower cost. This adjustment reduces the average path cost by 4.25%, 3.51%, 6.95%, and 4.87% across the four environments. The subsequent application of the KC-Smooth strategy smooths the corner regions of the path, and the total reductions in average path cost reach 6.94%, 5.40%, 8.16%, and 4.97% compared to the initial path. These results demonstrate that the optimization process effectively reduces path cost, thereby minimizing energy consumption in practical applications.

### 4.4. Evaluation of Smoothing Effects Under Kinematic Constraints

Trajectory smoothing is performed using the KC-Smooth strategy. Figure 13, Figure 14, Figure 15 and Figure 16 present the analysis of its impact on motion-level smoothness under kinematic constraints by illustrating the magnitude profiles of key kinematic parameters. Specifically, we evaluate whether the key kinematic parameters (velocity, acceleration, and jerk) satisfy the predefined constraint limits and exhibit temporal continuity throughout the trajectory. In the experiments, the constraint limits for velocity, acceleration, and jerk are all set to 50. In each figure, the gray dashed line represents the constraint threshold, while the colored solid lines indicate the magnitude profiles of the corresponding kinematic parameters.

After applying KC-Smooth, the magnitude profiles of velocity, acceleration, and jerk remain entirely within the specified constraint limits in all environments and show consistent temporal continuity. The velocity magnitude curves demonstrate smooth transitions and maintain near-constant speed within each segment. The acceleration profiles closely fit the expected curves of the adapted Jerk-Continuous S-Curve ACC/DEC algorithm, and the jerk profiles show no abrupt changes. These results indicate that KC-Smooth effectively achieves motion smoothness that meets the requirements for stable execution. By ensuring continuous transitions in motion, it reduces impact forces caused by sudden changes in acceleration, as well as flexible vibrations induced by jerk discontinuities. The resulting motion smoothness is conducive to stable system behavior and may help mitigate mechanical stress and extend operational longevity.

### 4.5. Discussion of Results

Based on the preceding comparative analysis, the proposed KSBB-P-RRT* demonstrates clear advantages over the other two algorithms in terms of planning time, path cost, and number of iterations. Its superior performance can be attributed to targeted improvements in the following aspects.

Fast Path Search Strategy: Building on Bi-RRT*, the DB-Sample strategy is introduced to dynamically adjust the sampling point at each iteration, balancing the breadth and depth of exploration. In addition, the AP-Steer strategy is incorporated to enable adaptive expansion and enhance obstacle avoidance capability. These combined improvements significantly enhance the adaptability of the algorithm to diverse environments, resulting in reduced planning time across all scenarios and a substantial decrease in the number of iterations, thereby achieving fast and efficient path search.MT-Optimize Strategy: Based on the triangle inequality principle, redundant waypoints in the initial path are removed, while the remaining waypoints are adjusted to yield paths that better conform to obstacle boundaries. This refinement effectively reduces the overall path cost across different environments.KC-Smooth Strategy: An adapted Jerk-Continuous S-Curve ACC/DEC algorithm is introduced for trajectory smoothing, which reduces path cost to some extent by refining corner transitions. At the same time, temporal kinematic constraints are satisfied, enabling the planned path to be directly executable and ensuring smooth motion control.

## 5. Conclusions

In this study, the KSBB-P-RRT* algorithm was proposed to address the challenges of robotic manipulator path planning under kinematic constraints. The algorithm achieves shorter planning time, lower path cost, and reduced number of iterations, making it suitable for fine-grained and precision-demanding applications. Its contributions can be summarized as follows. First, a fast path search strategy integrates a DB-Sample for improved sampling guidance with an AP-Steer for enhanced obstacle avoidance, thereby accelerating bidirectional exploration. Second, an MT-Optimize strategy removes redundant waypoints and adjusts remaining nodes based on the triangle inequality, effectively reducing path cost. Third, a KC-Smooth strategy adapts the Jerk-Continuous S-Curve ACC/DEC algorithm to generate continuous trajectory points, which are both spatially smooth and temporally executable, thus integrating path planning with motion control.

Across four benchmarks, KSBB-P-RRT* reduced planning time by 33–70% and path cost by 3–8% relative to Bi-RRT*, while requiring fewer iterations. Moreover, the generated trajectories satisfy kinematic constraints, supporting continuity and feasibility in tracking scenarios. These quantitative improvements demonstrate effectiveness of the proposed method in simulation and suggest its strong potential in tasks requiring smooth and reliable motion planning in complex environments.

Future work will focus on implementing the proposed KSBB-P-RRT* algorithm on real-time robotic platforms to further validate its applicability and effectiveness under real-world conditions. Moreover, since the algorithm primarily addresses end-effector trajectory planning in task space, the corresponding joint trajectories will need to be derived through inverse kinematics according to the specific manipulator configuration.

Despite the promising results, this study has several limitations. In particular, the current experiments are restricted to simulation scenarios with static obstacles, which limits the applicability of the method in dynamic and uncertain environments. Future research may also consider extending the algorithm to dynamic scenarios, including real-time path planning and obstacle avoidance.

## Figures and Tables

**Figure 1 sensors-25-05598-f001:**
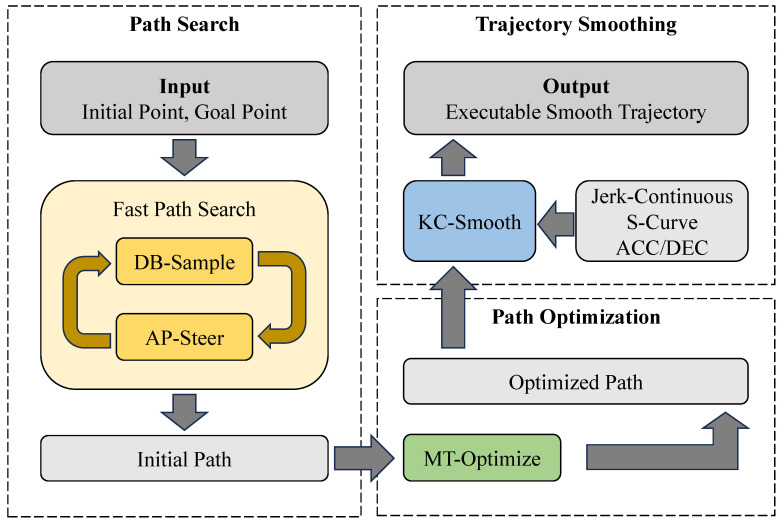
General framework of the proposed KSBB-P-RRT* algorithm.

**Figure 2 sensors-25-05598-f002:**
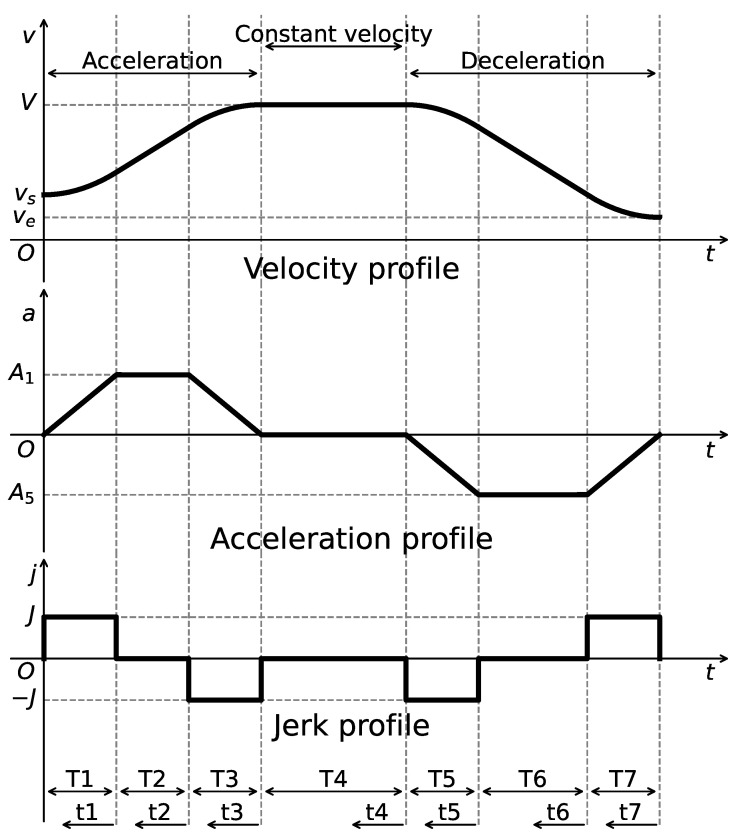
Velocity, acceleration, and jerk profiles of the S-Curve ACC/DEC algorithm.

**Figure 3 sensors-25-05598-f003:**
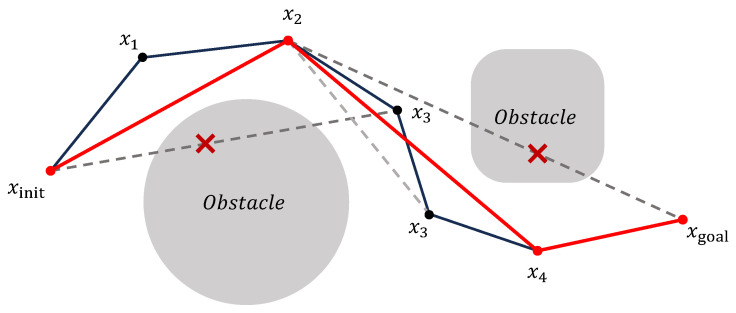
Illustration of the pruning process for eliminating redundant nodes in the initial path.

**Figure 4 sensors-25-05598-f004:**
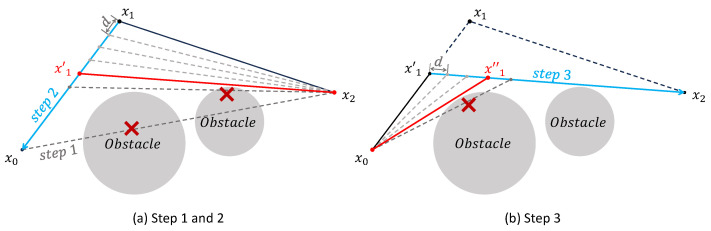
Illustration of the three-step path shortening strategy applied to a representative waypoint. Blue denotes the waypoint adjustment direction, and red the new path after fixation.

**Figure 5 sensors-25-05598-f005:**
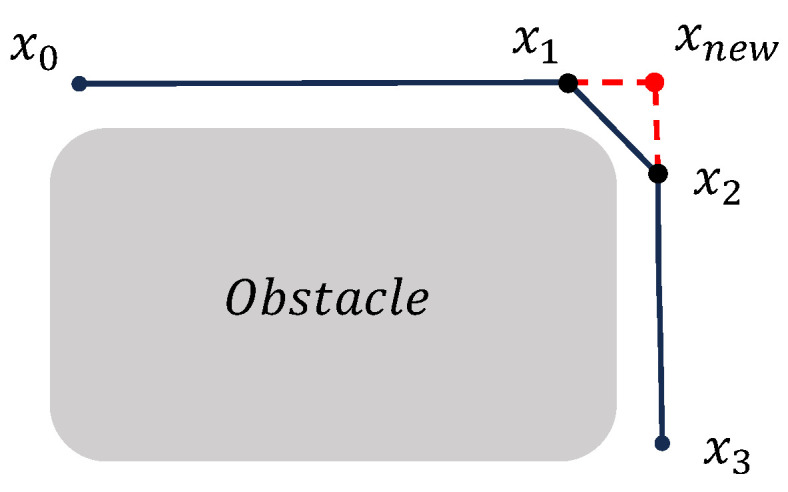
Illustration of the path repair strategy for handling overly close adjacent waypoints.

**Figure 6 sensors-25-05598-f006:**
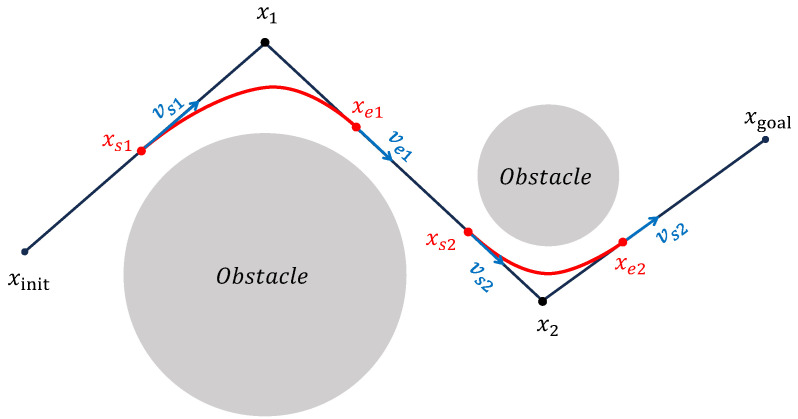
Illustration of corner smoothing under kinematic constraints for motion trajectory generation.

**Figure 7 sensors-25-05598-f007:**
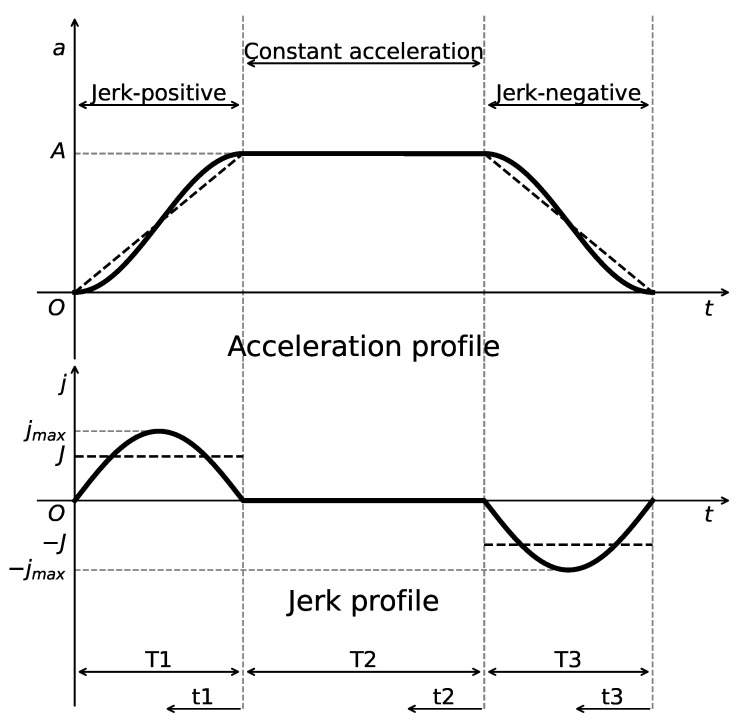
Acceleration and jerk profiles of the adapted Jerk-Continuous S-Curve ACC/DEC algorithm.

**Figure 8 sensors-25-05598-f008:**
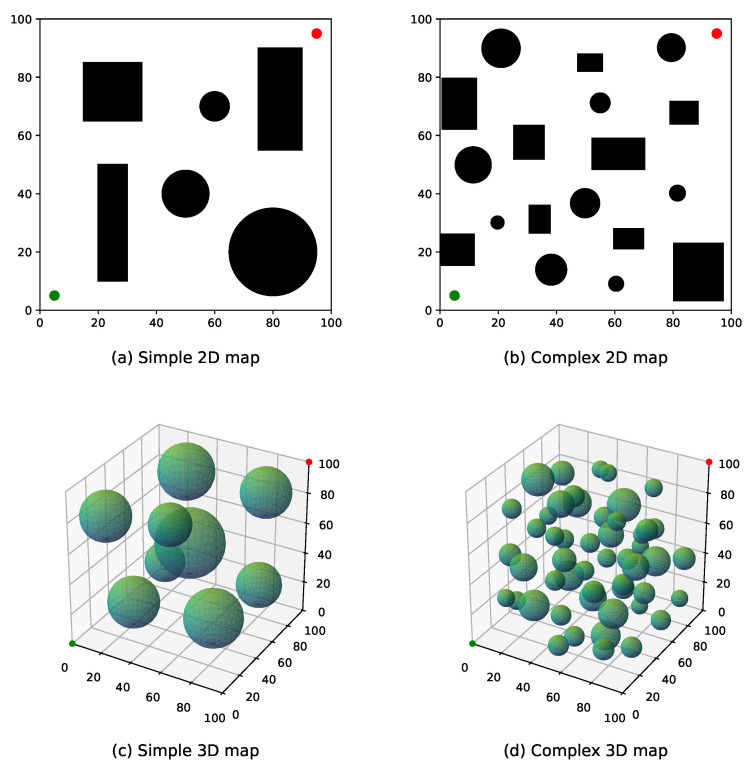
Test benchmark environments: (**a**,**b**) 2D maps and (**c**,**d**) 3D maps with varying obstacle complexity.

**Figure 9 sensors-25-05598-f009:**
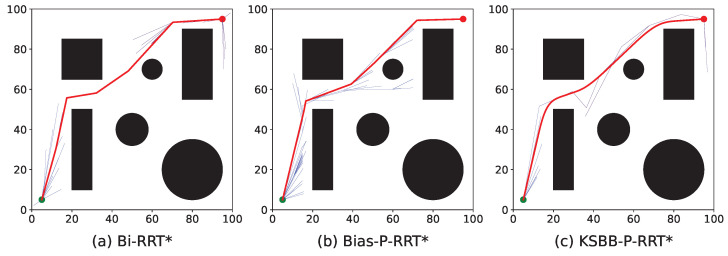
Simulation results of the three algorithms in the simple 2D environment.

**Figure 10 sensors-25-05598-f010:**
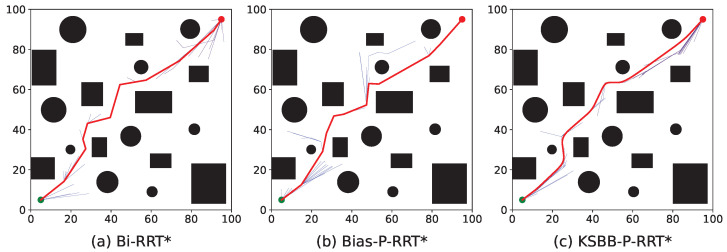
Simulation results of the three algorithms in the complex 2D environment.

**Figure 11 sensors-25-05598-f011:**
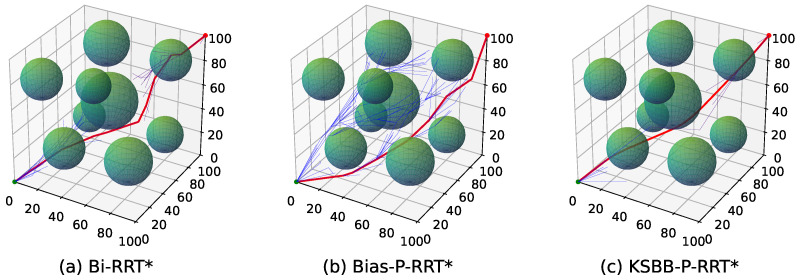
Simulation results of the three algorithms in the simple 3D environment.

**Figure 12 sensors-25-05598-f012:**
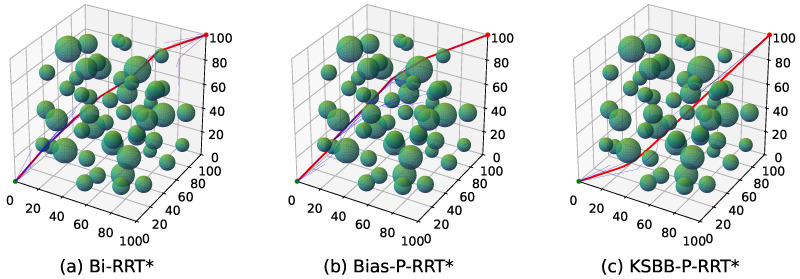
Simulation results of the three algorithms in the complex 3D environment.

**Figure 13 sensors-25-05598-f013:**
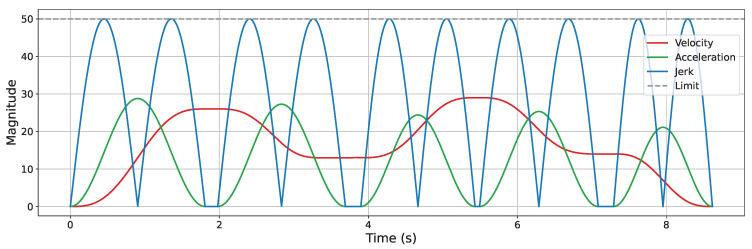
Velocity, acceleration, and jerk magnitude profiles in the simple 2D environment.

**Figure 14 sensors-25-05598-f014:**
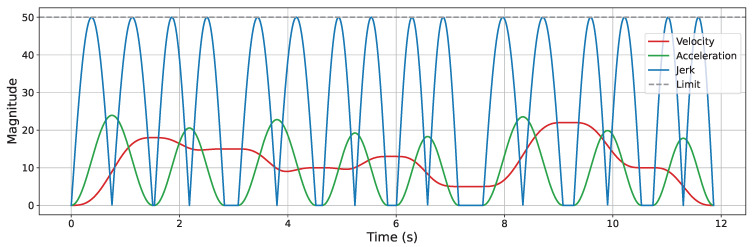
Velocity, acceleration, and jerk magnitude profiles in the complex 2D environment.

**Figure 15 sensors-25-05598-f015:**
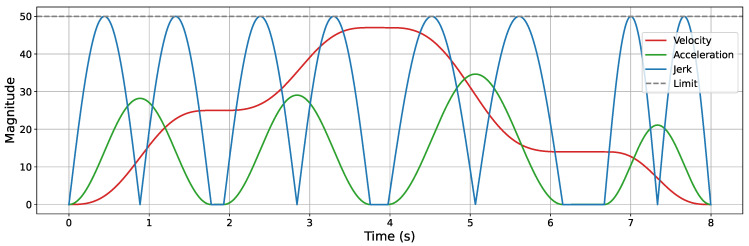
Velocity, acceleration, and jerk magnitude profiles in the simple 3D environment.

**Figure 16 sensors-25-05598-f016:**
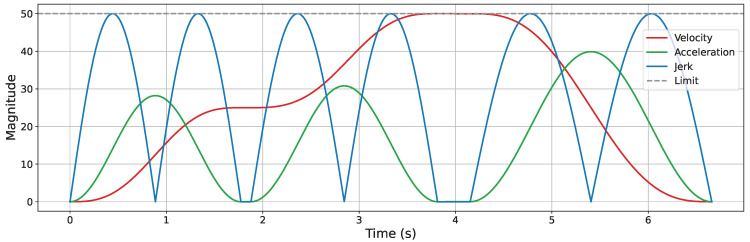
Velocity, acceleration, and jerk magnitude profiles in the complex 3D environment.

**Table 1 sensors-25-05598-t001:** Parameter settings of each algorithm.

Algorithm	$d_{\mathrm{step}}$	Iter	$N_{\mathrm{rewire}}$	$D_{\mathrm{obs}}$	$d_{\lambda }$	$p_{\mathrm{bias}}$
Bi-RRT*	5 (2D) / 7 (3D)	1500	10	–	–	–
Bias-P-RRT*	5 (2D) / 7 (3D)	1500	10	$\frac{1}{2}d_{\mathrm{step}}$	$\frac{1}{4}d_{\mathrm{step}}$	0.2
KSBB-P-RRT*	γ-adaptive (γ=15)	1500	10	$\frac{1}{2}d_{\mathrm{step}}$	$\frac{1}{4}d_{\mathrm{step}}$	[0, 0.6]

**Table 2 sensors-25-05598-t002:** Performance comparison of three RRT-based algorithms across four benchmark environments based on 100 experimental runs per case.

Environment	Algorithm	$T_{\mathrm{avg}}$ (s)	$C_{\mathrm{avg}}$	$I_{\mathrm{avg}}$	$I_{\max}$
Simple 2D	Bi-RRT*	0.338	151.109	107.49	224
	Bias-P-RRT*	0.497	146.665	172.20	430
	KSBB-P-RRT*	0.213	146.029	27.02	97
Complex 2D	Bi-RRT*	0.994	145.483	154.74	408
	Bias-P-RRT*	0.816	140.063	153.22	304
	KSBB-P-RRT*	0.663	136.153	61.74	252
Simple 3D	Bi-RRT*	0.524	210.003	297.21	1323
	Bias-P-RRT*	1.449	194.280	229.72	821
	KSBB-P-RRT*	0.231	192.976	15.84	28
Complex 3D	Bi-RRT*	2.276	189.668	59.59	140
	Bias-P-RRT*	2.084	179.051	55.96	291
	KSBB-P-RRT*	0.675	177.767	8.59	16

**Table 3 sensors-25-05598-t003:** Average path costs at three optimization phases for four benchmark environments based on 100 experimental runs per case.

Environment	$C_{\mathrm{init}}$	$C_{\mathrm{opt}}$	$C_{\mathrm{smooth}}$
Simple 2D	156.912	150.246	146.029
Complex 2D	143.925	138.874	136.153
Simple 3D	210.115	195.515	192.976
Complex 3D	187.065	177.960	177.767

## Data Availability

The raw data supporting the conclusions of this article will be made available by the authors upon request.

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
