# Peer review of "Fast Path Planning for Kinematic Smoothing of Robotic Manipulator Motion"

_sensors, 2025, doi:10.3390/s25175598_

Round 1

Reviewer 1 Report

Comments and Suggestions for Authors

It is a pleasure to review a demanding research topic, in the sense of using A Kinematically Smoothed Path Planning Algorithm for Smooth Motion of Robotic Manipulators. Upon Reading the paper, I came to observe the following issues, which I suggest improving its quality.

  1. Abstract: the performance of the suggested algorithm is explained in terms of time, path cost and iterations applied. To this effect, authors should also include the comparison results for the path cost as they have indicated for the time reduction. Furthermore, in line 5, it is stated that a novel algorithm is developed. However, this statement is unclear with the question that what makes it novel compared to the previous research works?
  1. In page 1(Introduction): the topic is written with lower case, and this issue including grammar and spelling should be checked throughout the document for consistency.
  2. In page 1(Introduction: the authors tried to introduce path planning and motion control, that is good starting point. However, they also drift to the application of path planning, where this is also a good idea. However, both ideas should be written in separate paragraphs instead of making them into one paragraph, and more support from literature is essential.
  3. Basic equations, figures, and algorithms are not supported with evidence(citations) or are not described if it is developed by the authors.
  4. In section 4(Simulation and Results): This part is well written and summarized. However the main contribution of the paper is missing, and it raises the question of what makes the research novel compared to the previous research works.
  5. Lastly, the organization of the paper is missing. Inclusion of the outline of the paper makes it easy for the reader to understand.

Thank you very much

Author Response

Thank you very much for your valuable comments and suggestions on our manuscript. We sincerely appreciate the time and effort you have taken to review our work. Please note that our detailed, point-by-point responses to all of your comments have been included in the uploaded file for your reference.

Reviewer 2 Report

Comments and Suggestions for Authors
  1. The title is long and needs to be rewritten in compact form. Furthermore, the title should be modified to not include any abbreviation.
  2. The abstract has to be revised and written in compact form. In the abstract part, the results and innovations of the study are not well clarified. In order to compare the results of the proposed method with previous traditional methods, it should be mentioned what quantitative criteria have been checked.
  3. The contribution of this paper is not clear. It suggests revising the contributions section and making these points clear and strong. The contribution has to be stated in points.
  4. Some abbreviations need to check when writing for the first time, like KD-RRT, APF-IRRT, and DBVSB-P-RRT.
  5. At the end of the introduction section, the author must add a paragraph that describes the paper organization.
  6. Some equations do not belong to the authors and I suggest to cite them, for example, Eq. (14).
  7. Not all symbols, variables and notation have been clearly defined.
  8. you may provide statistical results that validate the accuracy of the proposed method. i.e., provide the average resulted length and time of ten readings for each case.
  9. (The objective of this proposed method is to shorter planning time, lower path cost, and reduced number of iterations in the presence of obstacles). have you achieved this objective? discuss this point in your results, please.
  10. The cost function with respect to iteration has to be plotted. This an indication that the proposed algorithm works properly.
  11. A general block diagram has to be presented in the introduction part to visualize the general problem of this study.
  12. The conclusion is descriptive. It is void of quantitative and numerical improvement.
  13. It is interesting to implement the proposed method in real-time environment.
  14. Can these methods work with other types of robotic arms?
  15. There should be a further discussion about the limitation of the current works. To let readers better understand future work, please give specific research directions.
  16. However, there are still some issues and improvements to be addressed in future work. First, dynamic obstacles, unknown environment, obstacles’ shapes, and collision avoidance should be studied. In this paper, both the environment and obstacles are static relative to the robot, which is applicable in particular cases. In the future, it's very interesting to carry out using dynamic obstacles during the robot path planning process.

Author Response

(The authors gave the same response as above.)

Round 2

Reviewer 2 Report

Comments and Suggestions for Authors

I think that the authors have adequately addressed most of my comments in the revised version of the manuscript. Therefore, I have no further comments.